# Possibilities of Using Medicinal Plant Extracts and Salt-Containing Raw Materials from the Aral Region for Cosmetic Purposes

**DOI:** 10.3390/molecules27165122

**Published:** 2022-08-11

**Authors:** Izabela Nowak, Akmaral Issayeva, Marta Dąbrowska, Agata Wawrzyńczak, Henryk Jeleń, Bogusława Łęska, Azhar Abubakirova, Assel Tleukeyeva

**Affiliations:** 1Faculty of Chemistry, Adam Mickiewicz University, Uniwersytetu Poznańskiego 8, 61-614 Poznań, Poland; 2Ecology and Biology Research Institute, Shymkent University, Shymkent 160000, Kazakhstan; 3Faculty of Food Science and Nutrition, Poznań University of Life Sciences, Wojska Polskiego 31, 60-624 Poznań, Poland; 4M. Auezov South Kazakhstan University, Shymkent 160000, Kazakhstan

**Keywords:** Aral region, Zhaksy-Klych lake, bottom saline silt, medicinal plants, biochemical composition, cosmetic products

## Abstract

The aim of this work was to study the possibility of using medicinal plants in combination with salt-containing raw materials from the Aral Sea region for cosmetic purposes. The chemical and mineralogical compositions of salts occurring in this region were studied for pharmacological and cosmetic purposes. The salt-containing raw materials were studied by X-ray diffraction (XRD), thermogravimetric analysis (TGA) and scanning electron microscopy (SEM). The microflora of saline-containing raw materials and flora of the Zhaksy-Klych lake were studied. Fifty-six plant species were identified, of which 25% belong to the Asteraceae family, 32% were Poaceae, 22% were Amaranthaceae, and 21% were Tamaricaceae. Using the solid-phase microextraction (SPME) method and comprehensive two-dimensional gas chromatography–mass spectrometry, the composition of volatile compounds in such plant species as *Artemisia alba* L., *Achilleamillifolium* L., *Eleagus commutate Bernh. Ex Rydb*., *Psoraleadrupacea Bunge*, *Artemisia cipa O. Vegd*., *Thymus vulgaris* L., *Morus alba* L., *Salvia pratensis* L., *Glycyrhizaglabra* L., *Tanacetum vulgare* L., *Polygonumaviculare* L., *Alhagipseudoalhagi Gagnebin* and P*eganumharmala* L., chosen on the basis of their herbal components for future cosmetic products, was determined. In total, 187 different volatile compounds were found in the endemic plant species *Glycyrrhizaglabra* L., of which the following were dominant: 1,7-octadiene-3-, 2,6-dimethyl- with a peak area of 4.71%; caryophyllenes; bicyclo[7.2.0]4,11,11-trimethyl-8-methylene-, [1R-(1R*,4E,9S*)]—3.70%; bicyclo[2.2.1] heptane-2-1,7,7-trimethyl-,(1S)—3.46%; cyclohexanone, 5-methyl-2-(1-methylethyledene)-; 2-isopropyledene-5—2.97%; menthol; cyclohexanol, 5-methyl-2-(1-methylethyl)-; p-menthane-3-ol; menthol alcohol; and 2-isopropyl-5—2.08%. The remaining compounds were detected in amounts of less than 2.0%. Tests of seven cosmetic compositions developed on the basis of plant extracts and salt-containing raw materials revealed that three samples had a moisturizing effect. Launching the production of cosmetic products in the Aral region will not only reduce social tensions but also significantly improve the environmental situation in the region.

## 1. Introduction

The Aral region is associated with environmental problems such as desertification, salt storms and environmental pollution [1,2], but, in practice, this region is not considered to be a mineral resource base. The region is considered regressive in socio-economic terms: high unemployment, high morbidity and mortality, etc. The Aral Sea and the system of lakes located around it represent a mineral resource base of various salt-containing raw materials, ranging from various types of salts to different types of brine, silts and, importantly, waste after processing the initial salts [3]. Of particular interest is the salt lake Dzhaksy-Klych, which, as a result of drying, has turned into a salt deposit. It was revealed that the salt deposit of this lake is surrounded by a silt “pillow” on all sides (bottom and sides of the basin). The upper layer of halite is mainly underlain by a sulfate layer: astrakhanite, mirabilite, tenardite and, less often, others. The sulfate layer is underlain by a layer of silt. The bed of salt deposits is composed of dark brown clays and, less often, clay sands. Preliminary studies have shown the mineralogical characteristics of the coalite formation, represented by (in %): halite—90–96; epsomite—1.2–2.6; mirabilite—0.2–1.9; and gypsum—0.2–1.4 [4]. Using the methods of IR-Fourier spectroscopy, differential thermal analysis and differential scanning calorimetry, it was found that NaCl, accounting for 98.8 to 99.4%, dominates in the initial salt-containing raw materials of the Dzhaksy-Klych deposit, and inclusions of Na_2_CO_3_, CaSO_4_·2H_2_O, Na_2_SO_4_ and Na_2_SiO_4_ were noted. Some samples contain minerals with a more complex structure, such as astrakhanite (Na_2_Mg(SO_4_)_2_·4H_2_O). Some salt samples are mixtures of halite (NaCl), astrakhanite (Na_2_Mg(SO_4_)_2_-4H_2_O), magnesium sulfate hexahydride (MgSO_4_·6H_2_O), gypsum (CaSO_4_·2H_2_O) and sodium sulfate (Na_2_SO_4_), presumably in the form of mirabilite (Na_2_SO_4_·4H_2_O) [3]. In addition to halite salts, there are significant reserves of sulfate salts in the Dzhaksy-Klych deposit. It was found that the average capacity of the sulfate reservoir is 0.87 m in the Southern basin and 0.91 m in the Northern basin.

It is known that salt has antiseptic and whitening properties, relieves the skin of excess moisture and fat, promotes effective and gentle mechanical cleaning, and saturates the skin with minerals and macro- and microelements [5,6]. However, in describing the cosmetic effects of various salts, little attention has been paid to other salt-containing raw materials, such as brine, muddy sediments and shell rock. It should be mentioned that 85 to 90% of cosmetic products available in Kazakhstan are made from imported components, while domestic raw materials are used very little and inefficiently. The search for cosmetic ingredients and products that can preserve a youthful appearance and good skin condition for as long as possible is one of the urgent tasks of modern cosmetology [7].

Bioactive substances of natural origin have become especially useful in modern cosmetology. Biologically active compounds are those that have a special effect on the physiological and metabolic functions of the body [8,9,10,11]. These compounds are natural metabolites of plant processes, as well as organic and inorganic products of chemical transformations occurring in plant cells [12,13,14,15]. This is why plant extracts are very popular and are used in medicine, veterinary medicine, pharmacy, cosmetology and food production [16,17,18,19,20]. The effects of extracts of a number of medicinal plants have been described in the literature. Many flavonoids have antioxidant functions and medicinal value, e.g., ipriflavone, which has been reported to have a potential neuroprotective effect associated with its anti-inflammatory and antioxidant activity. Many medicinal mechanisms of flavonoids are closely related to their activity as radical scavengers. In previous studies, four flavonoids identified from sunflower flowers were known, including hymenoxin, sudaechitin, espidulin and pectolinarigenin [21,22,23].

The cosmetic market offers an increasing number of formulations containing plant extracts. Raw plant products are subject to standardization, which is a set of procedures aimed at ensuring a constant and reproducible level of biologically active components in each raw material division. Plant extracts are one of the most abundant and widely used cosmetic raw products of plant origin. Their widespread use and popularity are due to the richness of the biologically active components that they contain, which have a multidirectional effect on the skin.

Currently, out of more than 3000 drugs used in domestic medicine, 40% are produced from medicinal plants. Every year, their number increases. Medicinal plants are often preferred due to their low toxicity and the possibility of long-term use without side effects. According to [24], 342 species of vascular plants belonging to 43 families and 170 genera have been registered in the Aralkum desert in Kazakhstan, whose distribution is as follows: Chenopodiaceae (83 species), Asteraceae (45), Brassicaceae (32), Fabaceae (22), Roaseae (19), Boraginaceae (13), Suregaseae (5), Ariaseae (5). The flora of the Aral coast includes 414 species belonging to 43 families and 192 genera.

Thus, it should be noted that the Aral Sea region can be characterized not only as an ecological disaster zone but also as a mineral resource base rich in a variety of salt-containing and vegetable raw materials. The study of these raw materials as future components of cosmetic products can serve as a prerequisite for the development of the cosmetic industry in Kazakhstan. In this regard, the purpose of this study was to study the possibility of using medicinal plants in combination with salt-containing raw materials from the Aral Sea region for cosmetic purposes.

## 2. Results and Discussion

### Physicochemical Composition of Salts

Usually, all types of salts or mixtures are considered salt-containing raw materials, as in the case of the cosmetic use of Dead Sea salts. In the case of the studied mineral resource potential of Lake Dzhaksy-Klych, in addition to deposits of halite, sulfate and magnesium salts, salt-containing raw materials are represented by silty mud and rapa.

Five different samples, S1 (46°56′3.38″; 62°2′32.34″), S2 (46°55′53,09″; 62°3′29.09″), S3 (46°56′0,81″; 62°2′46.49″), S4 (46°55′55,66″; 62°3′14.93″) and S5 (46°56′7,71″; 62°3′16.93″), from Dzhaksy-Klych Lake were chosen for the study. Bottom silty mud, by origin and chemical composition, is categorized as the continental silt-sulfide type, and according to the results of physicochemical studies, it was found that its composition includes sand, clay, sulfurous iron compounds and colloidal substances of mineral and organic origin. The amount of water in the mud varies from 35–40 wt.%. The ionic composition of the mud liquid-phase solution is as follows: sodium—from 1.99 to 18.12 wt.%; sulfate ion—from 25.7 to 44.23 wt.%.; calcium—from 1.11 to 2.16 wt.%.; magnesium—from 3.89 to 4.24 wt.%.; potassium—from 0.78 to 1.11 wt.%.; carbonate ion—from 0.22 to 0.57 wt.%.; and chlorine—from 1.89 to 3.11 wt.%. In addition, it contains a large number of trace elements.

Microbiological examination showed the presence of mobile heterotrophic rod-shaped and coccoid bacteria in halite salt samples taken from depths of 0–10 cm. The largest number of bacteria, 10^3^ CFU/g, was found in samples collected along the coastline, while the number of bacteria decreased to 10–10^2^ CFU/g as the distance from the shore increased to 10–12 m.

XRD studies in the range of high 2Θ values allowed us to determine the mineral composition of the samples tested. Identification was performed on the basis of reference XRD spectra available in the PDF-4 + 2021 crystallographic database. The results presented in Figure 1a show that halite, the mineral form of NaCl, was present in the compositions of all samples. Furthermore, bloedite, i.e., Na_2_Mg(SO_4_)_2_·4H_2_O, was present in samples S2–S4, gypsum (CaSO_4_·2H_2_O) was found in samples S3–S5, and hexahydrite (MgSO_4_·6H_2_O in monoclinic form) was detected in samples S3 and S4. The most complex composition, which was found in sample S4, was characterized by the presence of mirabilite (Na_2_SO_4_·10H_2_O) in addition to the minerals mentioned above. The corresponding diffractograms from the database that confirm the assignment of individual reflections are shown in Figure 1b–f. Summarizing the results obtained from the XRD measurements, it can be concluded that the dominant minerals in the studied sediments are halite (typical reflections at the following 2Θ angles: 27.3°, 31.7°, 45.5° and 56.5°), present in all studied samples, and bloedite (2Θ angles: 19.5°, 20.7°, 26.1°, 27.1°, 27.4°, 30.1° and 44.7°) and gypsum (2Θ angles: 11.7°, 20.8°, 23.5°, 29.2°, 31.2° and 33.4°), which are present in three of the five studied samples. The data obtained are also summarized in Table 1 and agree well with previous studies [3,4,25], indicating the presence of the above-mentioned minerals in the Dzhaksy-Klych deposit.

Thermal analysis (TG/DTA) is able to provide a quick overview of water release, as well as dehydroxylation and decomposition of the samples as a function of rising temperatures. Figure 2 summarizes TG/DTA data for samples S3–S5.

The peak at ~100 °C corresponds to the loss of water molecules physically adsorbed on the surface of the samples. Additionally, the loss of hydration water from the hexahydrite molecule is observed in this temperature range, as seen for samples S3 and S4 (Figure 2a,b).

TG/DTA curves obtained for samples S3–S5 and presented in Figure 2a–c show two major peaks at temperatures higher than 100 °C, which are typical of thermally modified gypsum. They indicate two overlapping thermal processes: the first one at ~150 °C can be interpreted as the partial dehydration of gypsum, leading to calcium sulfate hemihydrate (basanite, CaSO_4_·0.5H_2_O) formation, whereas the second peak at ~180 °C may be attributed to the complete dehydration of gypsum, giving the anhydrite form (γ-CaSO_4_). A small exothermic effect present at 340 °C for sample S3 may be assigned to the phase transition of γ-CaSO_4_ into the β-form [25]. These observations prove the presence of gypsum in S3–S5 salt samples, which is in good agreement with the previously discussed XRD measurements.

According to [26], the dehydration of bloedite occurs stepwise, forming dihydrate Na_2_Mg(SO_4_)_2_·2H_2_O as an intermediate compound (~110 °C) and an anhydrous salt (~250 °C). Figure 2a,b show only the second dehydration step of bloedite at ~270 °C, since the first one overlaps the peak stemming from physically adsorbed water. However, for sample S4, a small shoulder at 108 °C can be observed, indicating that this sample contains higher amounts of bloedite than sample S3.

The peaks in the DTG curves occurring at temperatures ≥ 600 °C can be attributed to the solid–solid crystal phase transition and solid–liquid phase changes occurring, for example, for Na_2_SO_4_. Such transformations are confirmed by TG curves, which, in these temperature ranges, do not indicate mass loss for the analyzed samples S3 and S4.

Pure NaCl does not form hydrates above 0 °C; therefore, it does not decompose in the manner typical of other salts forming hydrates. Moreover, it does not show any mass loss up to almost 330 °C [27]. However, natural halite usually contains traces of gypsum, and thus, it behaves differently, since gypsum easily forms a dihydrate upon contact with water vapors. As a result, in the TG curve for sample S5 in Figure 2a, a small peak at 100 °C and an immense double peak at 149 °C and 176 °C can be observed, proving the release of adsorbed and crystallization water from gypsum, respectively.

Scanning Electron Microscopy (SEM) images provide much structural and morphological information on the materials studied. As one can see from the SEM images, halite occurs as subhedral to anhedral cubic crystals ranging from 20–70 μm in size (sample S1, Appendix A). SEM images of mirabilite show its subhedral crystal habit with a size of 4–15 μm (Appendix A shows the SEM image of sample S5 for comparison with the SEM of sample S1 in Appendix A). Mirabilite is the hydrated form of sodium sulfate and is more stable than the other minerals; thus, it was observed in all samples [28]. To establish the size distributions, around 100 particles were analyzed for each sample in SEM micrographs. First, the area—expressed as pixels in the digital image—occupied by a given particle was measured, and then the equivalent radius of the particle, assuming its spherical shape, was calculated by the ImageJ program. The results obtained are listed in Table 1, showing the distribution of particle sizes (histograms) and the mean particle size. The narrowest particle size distribution characterizes the sample with the most complex composition (S4), while the widest peak on the histogram can be observed for samples S2 and S5. On the other hand, samples S1 and S3 have similar peak widths in the histograms, but their maximums are shifted, which is manifested in the value of the mean particle size, reaching 51 ± 29 nm and 57 ± 22 nm for samples S1 and S3, respectively. The highest proportion of large particles characterizes sample S5, consisting of halite and gypsum, for which the mean particle size is 71 ± 33 nm, while the sample consisting only of halite is characterized primarily by the presence of particles with smaller sizes (mean particle size 51 ± 29 nm).

The measured pH of all samples was in the range of 6.79–7.335, while brine mineralization was 299.29–428.18 g/dm^3^. 

In the extracts of *Artemisia alba* L.*, Achillea millifolium* L.*, Eleagus commutate Bernh.ex Rydb., Psoralea drupacea Bunge., Artemisia cina O.Berg, Thymus vulgaris* L.*, Morus alba* L.*, Salvia pratensis* L.*, Glycyrhiza glabra* L.*, Tanacetum vulgare* L.*, Polygonum aviculare* L., *Alhagi pseudoalhagi Gagnebin. and Peganum harmala* L., the composition of volatile compounds was determined by solid-phase microextraction and complete (comprehensive) two-dimensional gas chromatography–mass spectrometry (SPME-GC × GC-ToFMS). The extraction technique allows for volatile preconcentration and, combined with the peak capacity of two-dimensional chromatography, results in the identification of numerous volatiles. For example, in the composition of the volatile compounds of *Artemisia alba* L. (Figure 3), the dominant among 176 compounds were (in %): Isoborneol—4,11; 2(3H)-furanone, 5-ethenyldihydro-5-methyl—2.72; 2(5H)-furanone, 5,5-dimethyl—2.74; 1-methylcycloheptanol—2.53; and bicyclo [2.2.1]heptan-2-one, 1,7,7-trimethyl-(1S)—2.53. 

The composition of volatile compounds in the extracts of *Achillea millifolium* L. (Figure 3) contained 191 volatiles, of which the following compounds were dominant: (in %): cyclohexanol, 5-methyl-2-(1-methylethyl)-, [1R-(1à,2á,5à)]—4.16; cyclopropanemethanol, 2,2,3,3-tetramethyl—3,45; caryophyllene—2.50; cyclohexanol, 1-methyl-4-(1-methylethyl)—2.29; and 3-cyclohexene-1-methanol, à,à,4-trimethyl-, acetate—1.97. The following compounds were found in amounts from 1.00 to 1.97%: 2-isopropyl-5-methyl-6-oxabicyclo [3.1.0]hexane-1-carboxaldehyde; 2-cyclohexen-1-one, 2-methyl-5-(1-methylethenyl)-; 2-cyclohexen-1-ol, 2-methyl-5-(1-methylethenyl)-, cis- 1,7-octadien-3-ol, 2,6-dimethyl-; 7-Oxabicyclo[4.1.0]heptane, 1-methyl-4-(2-methyloxiranyl)-; caryophyllene oxide; 7-oxabicyclo[4.1.0]heptan-2-one, 3-methyl-6-(1-methylethyl)-; nona-3,5-dien-2-ol; 3-hepten-2-one, 4-methyl-; bicyclo [2.2.1]heptan-2-one, 1,7,7-trimethyl-, (1S)-; 1,3,3,4-tetramethyl-2-oxabicyclo [2.2.0]hexane; 2,3-dioxabicyclo [2.2.2]oct-5-ene, 1-methyl-4-(1-methylethyl)-; 1-acetoxy-p-menth-3-one; borneol; 3-nonen-2-one; D-verbenone; 2-naphthalenemethanol, decahydro-à,à,4a-trimethyl-8-methylene-, [2R-(2à,4aà,8aá)]-; pulegone; cyclohexanone; and 5-methyl-2-(1-methylethyl).

*Eleagus commutate* Bernh. ex Rydb. (Appendix A) contained 194 volatile compounds, of which the following occurred with the highest contents: borneol; 2,7,7-trimethylbicyclo [2.2.1]heptan-2-ol—3.81%; caryophyllene—2.61%; 7-oxabicyclo [4.1.0]heptan-2-one, 3-methyl-6-(1-methylethyl)—2.46%; cyclohexanol, 5-methyl-2-(1-methylethyl)-, [1S-(1à,2à,5á)]—2,30%; and 2-cyclohexen-1-one, 2-methyl-5-(1-methylethenyl)-, (S)—2.09%. The following compounds were found in amounts from 1.00 to 1.87%: bicyclo [2.2.1]heptan-2-one; 1,7,7-trimethyl-; (1R)-, cyclohexanone, 5-methyl-2-(1-methylethyl)-; (2R-cis)-, 2(5H)-furanone; 5,5-dimethyl-, cyclohexanone; 5-methyl-2-(1-methylethyl)-, trans-3-nonen-2-one; caryophyllene oxide; methyltartronic acid; 1-pentanol; 2-dodecen-4-yne, (Z)-; terpineol, cis-á-; bicyclo [4.1.0]heptane, 3,7,7-trimethyl-, [1S-(1à,3á,6à)]-; hexanal; cyclohexanol, 5-methyl-2-(1-methylethyl)-, [1R-(1à,2á,5à)]-; and pentanal. 

*Artemisia cina O*. Berg is one of the endemic species of Southern Kazakhstan, but recent studies have shown that the plant species *Seriphidium cinum* [29], whose range extends to the borders with Iran, according to taxonomic characteristics, fully corresponds to the species *Artemisia cina O*. Berg. Nevertheless, the plant has long been considered narrowly endemic, with the area of occurrence confined to Kazakhstan. At the beginning of the 20th century, a santonin plant was built near Shymkent, which was engaged in the cultivation of A. cina and the manufacture of medicines from it. In this species, 190 volatile compounds were identified (Appendix A), dominated by (in %): á-myrcene—3.36; cyclohexanol, 5-methyl-2-(1-methylethyl)-, [1R-(1à,2á,5à)]—3.33; and caryophyllene—3.33. The contents of the following compounds were found to range from 2.04 to 2.42%: 1,6-cyclodecadiene, 1-methyl-5-methylene-8-(1-methylethyl)-, [s-(E,E)]-; cyclohexanone, 5-methyl-2-(1-methylethylidene)-; 7-oxabicyclo [4.1.0]heptane, 1-methyl-4-(2-methyloxiranyl)-; and 2-isopropenyl-5-methylhex-4-enal. 

*Thymus vulgaris* L. contained 134 volatile compounds in its biochemical composition (Appendix A), among which the following dominated (in %): santolina alcohol—4.42; methyl vinyl ketone—4.02. Mint furanone; 2,6-octadien-1-ol, 3,7-dimethyl-, propanoate, (E)-2(5H)-furanone, 5,5-dimethyl-; 5-hepten-2-one, 6-methyl-; and methyltartronic acid were revealed in amounts from 2.08 to 2.59%. The compounds 7-oxabicyclo[4.1.0]heptan-2-one, 3-methyl-6-(1-methylethyl)-; 7-oxabicyclo[4.1.0]heptan-2-one, 6-methyl-3-(1-methylethyl)-; 7-oxabicyclo[4.1.0]heptane, 1-methyl-4-(2-methyloxiranyl)-; hexanal; 3-caren-10-al; pulegone; acetic acid; bicyclo[3.1.1]hept-3-en-2-one, 4,6,6-trimethyl-; p-menth-1-en-8-ol; bicyclo[3.1.1]hept-2-en-6-one; and 2,7,7-trimethyl- occurred in quantities from 1.21 and to 1.89%. 

*Morus alba* L., a widespread plant in the southern regions of Kazakhstan, is often used for decorative purposes in the urban flora of cities. Its fruits have a moderate hypoglycemic effect, and decoction of its leaves exhibits anti-inflammatory, expectorant, antibacterial, astringent, diuretic and diaphoretic properties. The analysis of volatile compounds showed the presence of 198 compounds (Appendix A), of which the predominant were: cyclohexanol, 5-methyl-2-(1-methylethyl)-, [1R-(1à,2á,5à)]—6.35%; cyclohexanol, 1-methyl-4-(1-methylethyl)—4.32%; 2-cyclohexen-1-one, 2-methyl-5-(1-methylethenyl)-, (S)—3.07%; caryophyllene—2.72%; and 3-nonen-2-one—2.62%. 

*Alhagi pseudoalhagi* Gagnebin is a plant that occurs in some sites in the semi-desert territories of both the Kyzyl-Orda and Turkestan regions, forming carpets. In folk medicine, it is used to treat colds and diseases of the gastrointestinal tract and as an astringent, hemostatic, choleretic, and wound healing and bactericidal agents. Of the 195 identified volatile compounds (Appendix A), bicyclo [3.1.1]hept-3-en-2-ol, 4,6,6-trimethyl-; isoaromadendrene epoxide; and 1-oxetan-2-one, 4,4-diethyl-3-methylene were dominant, and their content was 3.01, 2.91 and 2.28, respectively. The following compounds occurred in quantities from 1.00 to 1.95%: caryophyllene oxide, 7-oxabicyclo [4.1.0]heptane, 1-methyl-4-(2-methyloxiranyl); 2,6-Octadien-1-ol, 3,7-dimethyl-, (Z)-; benzenemethanol, à,à,4-trimethyl-; bicyclo[2.2.1]heptan-2-one, 1,7,7-trimethyl, (1S)-; 2-cyclohexen-1-one, 2-methyl-5-(1-methylethenyl)-; 7-oxabicyclo[4.1.0]heptan-2-one, 3-methyl-6-(1-methylethyl)-; ethanone, 1-(1-cyclohexen-1-yl)-; 5-hepten-2-one, 6-methyl-; borneol; bicyclo[3.1.1]hept-3-en-2-one, 4,6,6-trimethyl-, (1S)-; p-menth-1-en-8-ol; 1H-3a,7-methanoazulene, octahydro-1,4,9,9-tetramethyl-; Naphthalene, 1,2,3,4-tetrahydro-1,6-dimethyl-4-(1-methylethyl)-, (1S-cis)-.

*Peganum harmala* L., a widespread herbaceous plant, is a bioindicator of overgrazing. In Kazakh folk medicine, it is used as a bactericidal agent. The anti-inflammatory and antioxidant properties of the plant are described in [30], and the toxic effect of its seeds is revealed in [31]. In the developed cosmetic compositions, the extract of this plant acts as a bactericidal component. Analysis of the composition showed the presence of 187 volatile compounds (Appendix A), dominated by the following (in %): 2-cyclohexen-1-one, 2-methyl-5-(1-methylethenyl)-, (S)—5,32%; cyclohexanol, 5-methyl-2-(1-methylethyl)-, (1à,2á,5á)—3,78%; and 7-oxabicyclo[4.1.0]heptan-2-one, 3-methyl-6-(1-methylethyl)—3,43%. The contents of caryophyllene and 7-oxabicyclo[4.1.0]heptan-2-one, 3-methyl-6-(1-methylethyl) were 3.17 and 3.03, respectively. 

*Glycyrhiza glabra* L. licorice is a perennial plant growing in suburban areas of Shymkent. The plant is used in phytotherapy for skin diseases and has both anti-inflammatory and anti-ulcer activities, which is important because most anti-inflammatory drugs are ulcerative. The roots contain glycyrrhizin, the main water-soluble component, which is 50 times sweeter than sugar, 2-β-glucuronosylglucuronic acid and isoliquiritigenin-4-glucoside [32]. Glycyrrhizin is a non-hemolytic saponin showing foaming ability and is one of the most powerful absorbers of hydroxyl radicals. Of the 187 volatile compounds (Appendix A) found in the plant’s green phytomass, the following had the highest % content: 1,7-octadien-3-ol, 2,6-dimethyl-; caryophyllene; and bicyclo[2.2.1]heptan-2-one, 1,7,7-trimethyl-, (1S)-, i.e., 4.71, 3.69 and 3.46, respectively. 

*Tanacetum vulgare* L. is a plant that has long been used for the treatment of parasitic worms, gout and digestive disorders. The foliage of the plants has been used as a repellent. As a result of the conducted analyses, 206 volatile compounds were identified (Appendix A). The following compounds occurred in quantities from 1.02 to 1.85%: cyclohexanone, 5-methyl-2-(1-methylethylidene)-; borneol; 2-cyclohexen-1-one, 2-methyl-5-(1-methylethenyl)-, (S)-; (E)-3(10)-caren-4-ol; cyclohexanone, 5-methyl-2-(1-methylethyl)-; nona-3,5-dien-2-ol; 3-Penten-1-ol, 2,2,4-trimethyl-; bicyclo[2.2.1]heptan-2-one, 1,7,7-trimethyl-, (1S)-; bicyclo[3.1.0]hexan-2-ol, 2-methyl-5-(1-methylethyl)-, (1à,2á,5à)-; spiro [2.4]heptane-5-methanol, 5-hydroxy-; caryophyllene oxide; 3-cyclohexene-1-methanol, à,à4-trimethyl-; cyclohexanol, 5-methyl-2-(1-methylethyl)-, [1R-(1à,2á,5à)]-; naphthalene, 1,2,3,4,4a,5,6,8a-octahydro-7-methyl-4-methylene-1-(1-methylethyl)-, (1à,4aà,8aà)-; acetic acid, 1,7,7-trimethyl-bicyclo[2.2.1]hept-2-yl ester; cyclohexanol, 5-methyl-2-(1-methylethyl)-, [1R-(1à,2á,5à)]-; cyclotridecane; 1,3,3,4-tetramethyl-2-oxabicyclo [2.2.0]hexane.

*Polygonum aviculare* L. is a ubiquitous weed plant. In dermatology, the extract of this plant is used for the treatment of allergic skin diseases (eczema, atopic dermatitis, acne, boils, etc.), psoriasis, dermatomyositis, vasculitis, congenital epidermolysis, etc. In this species, 185 volatile compounds were found (Appendix A). In the phytomass of this plant, cyclohexanol, 5-methyl-2-(1-methylethyl)-, (1à,2à,5á) occurred in the amount of 4.6392%. Pulegone and 2-cyclohexen-1-one, 2-methyl-5-(1-methylethenyl)-, (S)- were found in the amounts of 2.33 and 2.15%, respectively.

Seven cosmetic compositions were developed on the basis of mixtures of halite, sulfate and magnesium salts of Lake Dzhaksy-Klych, as well as the extracts and powders of medicinal plants.

Analysis of the results obtained in the course of application tests was preceded by the determination of the expected effects of the tested cosmetic products: (a) decreases in SEr, SEsm, SEsc and SEw parameters; (b) an increase in epidermal hydration, where the most desirable results are values above 45 CM with a simultaneous decrease in TEWL levels, indicating the proper conditions for a skin barrier function.

Changes in epidermal hydration and associated changes in the TEWL level and in skin topography parameters are shown by the parameters collected in Table 2. The analysis of the obtained results concerning scrubs indicated a strong exfoliating effect of scrub 1. A significant increase in the value of transepidermal water loss and a decrease in epidermal hydration were observed after the application of this scrub, and such effects are expected of mechanical peels. Its strong effect was also confirmed by SELS parameters: a reduction in skin roughness (SEr and SEsc) with a simultaneous increase in the SEw value, associated with an intensive exfoliating effect as a scrub.

Considering the results obtained for scrub 2, intended for sensitive skin, the effects expected for an exfoliating product were not observed. The peeling effect of the product was negligible, manifested as a slight worsening of the TEWL value, and a moisturizing effect was even observed. It can, therefore, be concluded that scrub 2 is potentially suitable for extremely sensitive skin that cannot tolerate the use of intensive mechanical peels.

The opposite results were observed for scrub 3 (Figure 4), with a significant increase in TEWL (as expected after the application of a scrub) and the simultaneous worsening of skin hydration due to the water imbalance in the area of the superficial layers of the epidermis. In addition, a favorable decrease in SEsc and a smoothing effect on skin wrinkles (SEw), desired for cosmetic products designed for skin with scar-like problems, were noted. The last exfoliating product—scrub 7—showed a similar influence on TEWL and skin hydration to scrub 3 but did not lead to beneficial changes in terms of reducing the desquamation and roughness of the skin. It can be concluded that it did not fully fulfill its expected exfoliating function. As to the analysis of results obtained for cosmetic masks, for all preparations tested, the desired favorable moisturizing effect was observed: i.e., TEWL decreased with a simultaneous increase in the epidermal hydration value.

Nevertheless, the most beneficial effects were observed for universal mask 4 (Figure 3), whose application led to an improvement in SELS parameters, evening the skin tone (SEsm), reducing roughness and desquamation (SEr and SEsc) and showing a skin smoothing effect (SEsm and SEw).

Similar observations were made for mask 6, except that it had an ambiguous effect on the SEsc parameter, causing it to worsen, which, however, taking into account the ingredients used in its production, could be caused by the residue of the product remaining on the skin surface after the application, distorting the objective measurement. Mask 5, on the other hand, did not sufficiently reduce roughness (possibly due to its formulation being intended for sensitive skin) but nevertheless had a positive smoothing effect on skin tone and significantly reduced the level of excessive water loss from the epidermis—an effect that is highly desirable for products for hypersensitive skin. Other images of skin topography obtained before and after the application of the tested cosmetic products (1, 2 and 5–7), together with the values of the SELS parameters, can be found in the Appendix A.

## 3. Materials and Methods

### 3.1. Materials

The objects of the study were halite, sulfate, magnesium, mixed salts, brine and silt collected from the deposit of Dzhaksy-Klych (Aral district, Kyzylorda region); medicinal plants were collected in the territory of the Turkestan region.

### 3.2. Physicochemical Characterization of Salts

The salt-containing raw materials were studied by X-ray diffraction at high 2θ angles. The measurements were performed on a Bruker AXS D8 Advance diffractometer equipped with a Johansson monochromator and a LynxEye strip detector (Bruker; Billerica, MA, USA). CuKα radiation with a wavelength of λ = 0.154 nm was used. The range of 2θ was 6–60° with a step of 0.05°.

Thermogravimetric analysis was carried out on a Setsys 1200 Setaram instrument (Setaram instrumentation, Caluire-et-Cuire, France) in an air atmosphere. The final measurement temperature was 1000 °C, and the heating rate was 10°/min.

Scanning electron microscopy (SEM) was performed on a Quanta FEG 250 (FEI)(Mega-NK, Moscow, Russia) microscope under low-pressure vacuum (70 Pa) and 10kV beam accelerating voltage. Soil samples were placed on a carbon-coated copper grid (400-mesh). The obtained images were processed using the image processing software ImageJ SmartSCAN (NIH, Bethesda, MD, USA), which is a freely available, open-source, multithreaded platform. The results are presented as histograms with 255 gray levels.

Geological exploration of salt reserves was carried out in the fields of Buga-Dzhaily (Suzak district, Turkestan region), Dzhaksy-Klych and the Small Aral (near the city of Aralsk, Kyzyl-Orda region).

### 3.3. Plant Characterization

Taxonomic analysis of plant species was carried out using an illustrated herbarium for the identification of species in six volumes of “Flora of Kazakhstan, 1969” [33]. Harvesting, drying and quality assessment of medicinal plant raw materials were carried out in accordance with the requirements of the GOST 24027.1–80 standard (Medicinal plant raw materials. Methods for determining authenticity, infestation with barn pests. Grinding and impurity content).

The study of volatile compounds in the dried medicinal plants was carried out using solid-phase microextraction for their isolation and comprehensive two-dimensional gas chromatography–mass spectrometry for subsequent resolution and identification (SPME-GC × GC-ToFMS). For SPME extraction, PDMS fiber was used. Compounds were extracted from a 1g dry sample placed in a 20 mL headspace vial for 20 min. at 50 °C. Analyses were performed on an Agilent 6890N gas chromatograph coupled to a Pegasus 4 time-of-flight mass spectrometer (LECO, St. Joseph, MI, USA). The instrument was equipped with a CTC autosampler allowing liquid injection as well as SPME. Analyses were performed on a conventional column setup: the first column was a polar one (SLB-5, 30 m × 0.25 mm × 0.25 µm, Supelco, Bellefonte, PA, USA), and the second was Supelcowax-10 (0.91 m × 0.20 mm × 0.20 µm, Supelco, Bellefonte, PA, USA). Samples were injected (at 250 °C) in splitless mode. The oven temperature was programmed at 40 °C for 1 min., followed by 8 °C/min to 235 °C for the first oven, and 55 °C for 1 min., followed by 8 °C/min to 250 °C for the second oven. The modulator temperature was +15 °C compared to the oven temperature. The modulator period was 3 s with 0.6 s for the hot pulse and 0.9 s for the cold pulse. The mass spectrometer was operated in EI (electron impact) mode in a mass range of m/z 33–333 with a spectra acquisition speed of 150 full spectra/sec. Tentative identification of compounds was based on the comparison of their mass spectra with those in the NIST 02 mass spectral library. LECO ChromaTOF v. 4.50.8.0 was used for data processing. Volatile compound analysis was performed at the Poznan University of Life Sciences.

### 3.4. Cosmetics Production and Characterization

The production of cosmetic products from natural salt-containing and vegetable raw materials was carried out in accordance with the specifications of the Interstate standard GOST 31698-2013 [34].

The evaluation of the efficacy of seven cosmetic products (4 scrubs and 3 masks) was carried out in the laboratory of the Department of Applied Chemistry of Faculty of Chemistry of Adam Mickiewicz University (Poznań, Poland). The testing panel consisted of male and female volunteers of various ages and with different skin conditions. The in vivo study was aimed at determining the effect of a single application of cosmetic products 1–7 on the inner part of the left forearm. Evaluation of the skin condition using non-invasive methods included the use of the following devices: Tewameter^®^ TM 300 (to determine the level of transepidermal water loss, TEWL), Corneometer^®^ CM 825 (skin hydration) and Visioscan^®^ VC 98 (skin topography parameters: SEr—roughness; SEsm—smoothness; SEsc—scaliness; and SEw—wrinkles).

## 4. Summary and Conclusions

The optimal compositions of cosmetic products based on salt-containing and plant raw materials were selected on the basis of X-ray diffraction and thermogravimetric analyses. The initial salt-containing raw materials from the Dzhaksy-Klych deposit were found to contain NaCl contents ranging from 98.8 to 99.4%, in addition to Na_2_CO_3_, CaSO_4_·2H_2_O, Na_2_SO_4_, Na_2_SiO4. The mineralogical compositions of all samples studied are presented in Table 3.

The plants *Artemisia alba* L., *Achillea millifolium* L., *Eleagus commutate Bernh. ex Rydb.*, *Artemisia sipa O. Vegd.*, *Thymus vulgaris* L., *Morus alba* L., *Glycyrhiza glabra* L., *Tanacetum vulgare* L., *Polygonum aviculare* L., *Alhagi pseudoalhagi Gagnebin.*, *Peganum harmala* L., *Mentha arvensis* L. and *Melissa Officinalis* L., which formed the basis of the plant components of the cosmetic products, were subjected to solid-phase microextraction and complex two-dimensional gas chromatography–mass spectrometry (SPME-GCxGC- ToFMS). The determination of their biochemical compositions permitted the identification of 196 to 207 types of volatile compounds. Analyses of the topography of the skin, the hydration of the epidermis and transepidermal water loss as a result of the influence of mask and scrub application showed that this complex has a clear advantage over similar products and can be recommended for use in daily practice in beauty salons.

## Figures and Tables

**Figure 1 molecules-27-05122-f001:**
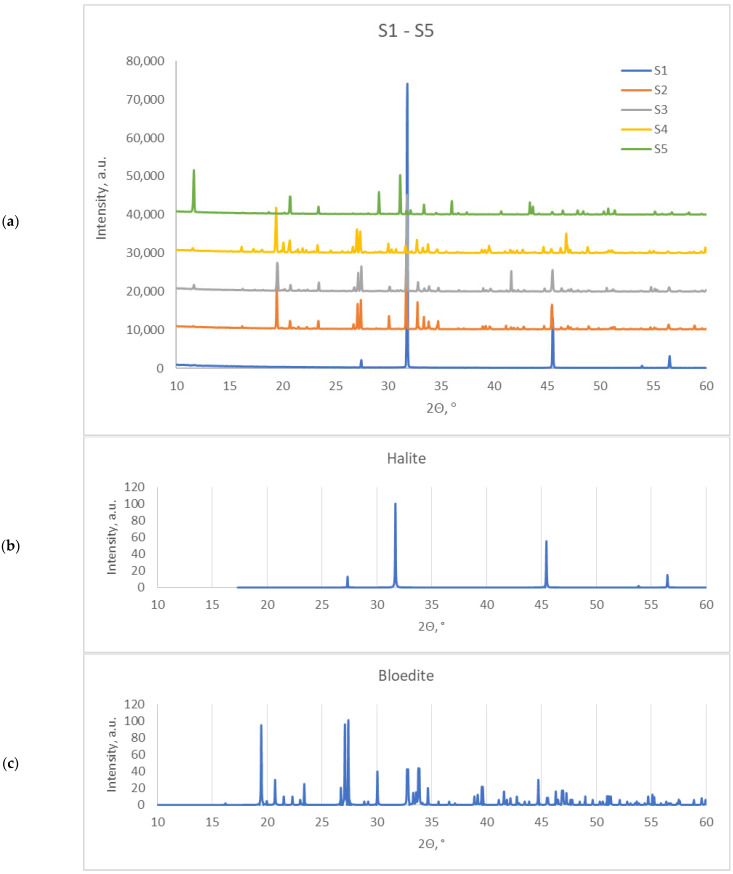
XRD diffractograms of the (**a**) S1-S5 salts and (**b**–**f**) respective XRD patterns from the crystallographic database.

**Figure 2 molecules-27-05122-f002:**
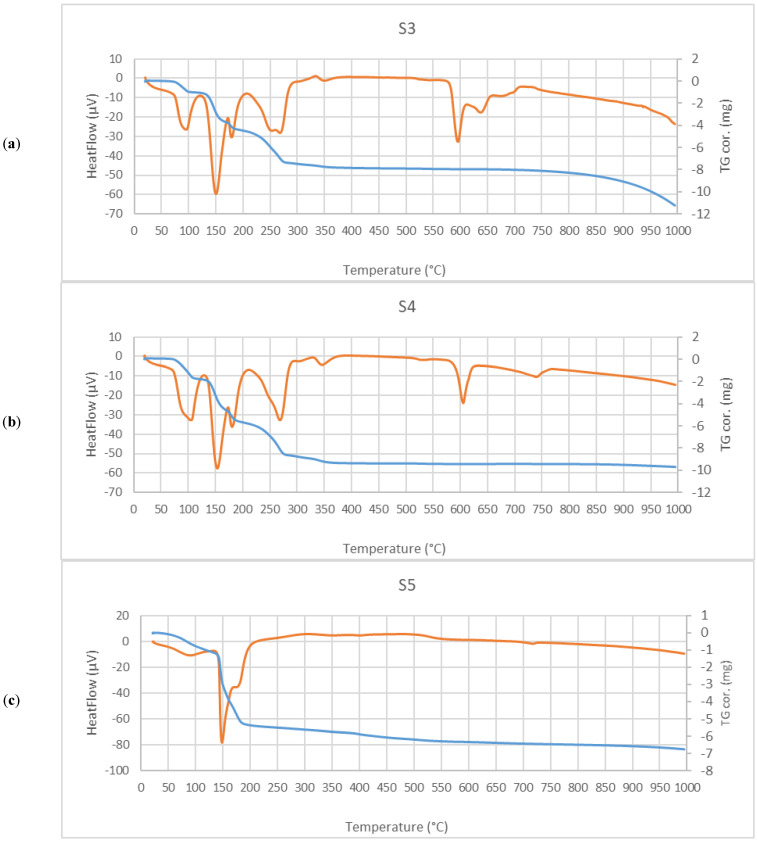
(**a**–**c**) Thermogravimetric data for samples S3–S5.

**Figure 3 molecules-27-05122-f003:**
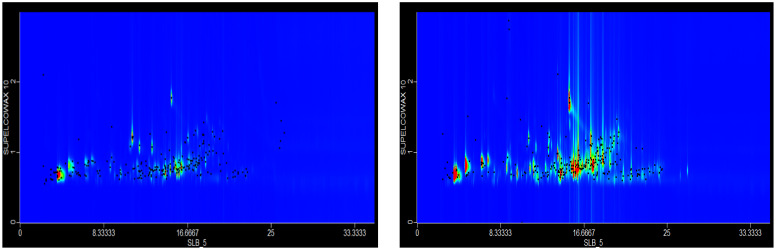
Chromatograms of *Artemisia alba* L. (**Left**) and *Achillea millifolium* L. (**Right**) obtained by SPME-GCxGCToFMS and used in cosmetic formulae. The remaining chromatograms of plants analyzed are provided in Appendix A.

**Figure 4 molecules-27-05122-f004:**
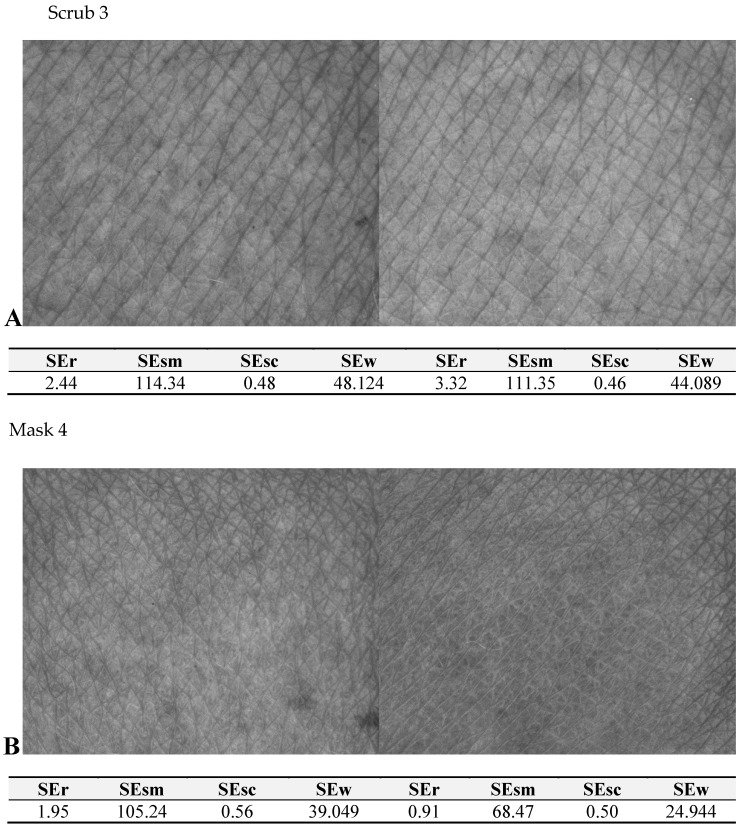
Skin topography images obtained before (**A**) and after application (**B**) of scrub 3 and mask 4 with the corresponding parameters of skin topography.

**Table 1 molecules-27-05122-t001:** Particle size distribution and mean particle size obtained from histograms.

Sample	Histogram of Particle Sizes (Gray Shade Distribution from ImageJ)	Mean Particle Size [nm]
S1	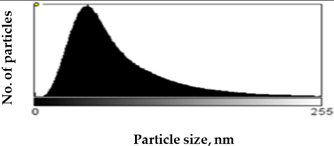	51 ± 29
S2	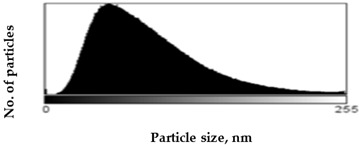	64 ± 33
S3	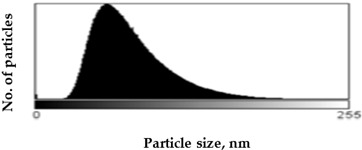	57 ± 22
S4	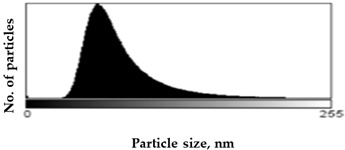	56 ± 22
S5	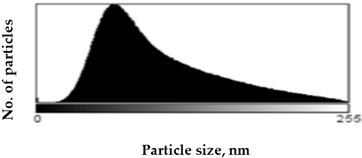	71 ± 33

**Table 2 molecules-27-05122-t002:** Changes in the basic skin parameters (skin hydration and TEWL levels) and skin topography parameters determined for cosmetic products 1–7 in application tests (data are expressed as percentage change ∆% ± 0.5%).

	TEWL [g/m^2^/hour]	HYDRATION [CM]	SEr [AU]	SEsm [AU]	SEsc [AU]	SEw [AU]
SCRUB 1	22.0	−13.4	−15.9	24.3	−13.8	36.7
SCRUB 2	1.6	13.4	3.8	14.8	21.6	15.9
SCRUB 3	−19.8	−14.5	36.1	−2.4	−4.2	−8.4
MASK 4	−24.8	16.9	−53.3	−34.9	−10.7	−36.1
MASK 5	−31.0	1.0	24.2	1.7	−9.7	−10.9
MASK 6	−10.2	7.4	−28.0	−5.9	28.3	−4.4
SCRUB 7	−32.6	−16.2	39.7	−3.9	42.1	−2.9

**Table 3 molecules-27-05122-t003:** Summary of the XRD, TG and SEM studies.

Sample	Halite	Bloedite	Hexahydrate	Gypsum	Mirabilite
S1	X				
S2	X	X			
S3	X	X	X	X	
S4	X	X	X	X	X
S5	X			X	

## Data Availability

The data presented in this study are available upon request from the authors.

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
