# Peer review of "Possibilities of Using Medicinal Plant Extracts and Salt-Containing Raw Materials from the Aral Region for Cosmetic Purposes"

_molecules, 2022, doi:10.3390/molecules27165122_

Round 1

Reviewer 1 Report

The paper focusses on the Possibilities of Using Medicinal Plants Extracts and SaltContaining Raw Materials from the Aral Region for Cosmetic
Purposes. The authors has given the vivid description of the materials & methods used in the study. The topic is quite novel & not much work has been made on this topic.

Author Response

Dear Reviewer, thank you for your valuable remarks. Additionally, the manuscript has been modified according to other reviewers suggestions. All changes are marked red in the body text of the article.

Reviewer 2 Report

I read the article about Possibilities of Using Medicinal Plants Extracts and SaltContaining Raw Materials from the Aral Region for Cosmetic Purposes. I found this article interesting and the fellow researchers in the same field can take advantages from it however it needs some modification.

The abstract is very poorly presented and it seems like the abstract of a review article. There is no need of background of the selected topic. It must be included with

aim, objectives, methodology, results, conclusion and possible future perpectives.

There is no statistical or mathematicla data in the abstract.

The introduction part seems insufficient. The sentences are started with some non-scientific wordings.

Description and discussion about figures 1 must be comprehensive and inteligible.

Table 1 is also confusing.

Overall there are numerous typos and grammatical errors.

Author Response

Dear Reviewer, thank you for your valuable remarks. The manuscript has been modified according to your suggestions. All changes (besides the English corrections for the clarity) are shown below and marked red in the body text of the article.

The abstract is very poorly presented and it seems like the abstract of a review article. There is no need of background of the selected topic. It must be included with aim, objectives, methodology, results, conclusion and possible future perpectives. There is no statistical or mathematicla data in the abstract.

Response: The abstract was rewritten The analytical data/methodology, etc.  have been added.

The introduction part seems insufficient. The sentences are started with some non-scientific wordings.

Response: The introduction has been revised by a professional English language interpreteur.

Description and discussion out figures 1 must be comprehensive and inteligible.

Response: The description of Figure 1 has been changed. A discussion of the results from XRD measurements has been supplemented with the results from the database and compared with the literature.

The revised excerpt from the paper is presented as follows:

XRD studies in the range of high 2Θ values allowed us to determine the mineral composition of the samples tested. Identification was performed on the basis of reference XRD spectra available in the PDF-4+ 2021 crystallographic database. The results presented in Figure 1a showed that halite, the mineral form of NaCl, was present in the compositions of all samples. Furthermore, bloedite, i.e., Na2Mg(SO4)2·4H2O, was present in samples S2-S4, gypsum (CaSO4·2H2O) in samples S3-S5, and also hexahydrite (MgSO4·6H2O in monoclinic form) in samples S3 and S4. The most complex composition of sample S4 was characterized by the presence of mirabilite (Na2SO4·10H2O) in addition to the minerals mentioned above. The corresponding diffractograms from the database that confirm the assignment of individual reflections are shown in Figures 1b-1f. Summarizing the results obtained from the XRD measurements, it can be concluded that the dominant minerals in the studied sediments are halite (typical reflections at the following 2Θ angles: 27.3 °; 31.7 °; 45.5 °; 56.5 °), present in all studied samples, bloedite (2Θ angles: 19.5 °; 20.7 °; 26.1 °; 27.1 °; 27.4 °; 30.1 °; 44.7 °) and gypsum (2Θ angles: 11.7 °; 20.8 °; 23.5 °; 29.2 °; 31.2 °; 33.4 °), which are present in three of the five studied samples. The data obtained are also summarized in Table 3 and agree well with previous studies [3,4, 25] indicating the presence of the above-mentioned minerals in the Dzhaksy-Klych deposit.

Table 1 is also confusing.

Response: Table 1 has been modified, and the data has been described in text in a more detailed form. The revised excerpt from the paper is presented as follows:

To establish the size distributions, around 100 particles were analyzed for each sample in SEM micrographs. First, the area – expressed as pixels in the digital image – occupied by a given particle was measured and then the equivalent radius of the particle, assuming its spherical shape, was calculated by the ImageJ program. The results obtained are listed in Table 1, showing the distribution of particle sizes (histograms) and the mean particle size. The narrowest particle size distribution characterizes the sample with the most complex composition (S4), while the widest peak on the histogram can be observed for samples S2 and S5. On the other hand, samples S1 and S3 have similar peak widths in the histograms, but their maximums are shifted, which is manifested in the value of the mean particle size, reaching 51±29 nm and 57±22 nm for samples S1 and S3, respectively. The highest proportion of large particles characterizes sample S5, consisting of halite and gypsum, for which the mean particle size is 71±33 nm, while the sample consisting only of halite is characterized primarily by the presence of particles with smaller sizes (mean particle size 51±29 nm).

Reviewer 3 Report

The current paper is very poorly constructed and prepared.

The plant names must be thoroughly revised. Italics must be used for plant names.

In the Materials and Methods section, a more detailed descriptions are necessary.

Both an Informed Consent Statement and an Institutional Review Board Statement are necessary due to the involvement of humans in the current study.

English language and style checking are required

Before a work can be considered for publication, the authors must meticulously prepare it.

Author Response

Dear Reviewer, thank you for your valuable remarks. The manuscript has been modified according to your suggestions. All changes are shown below and marked red (besides the English changes for the clarity) in the body text of the article.

The current paper is very poorly constructed and prepared.

Responce: The paper content has been improved.

The plant names must be thoroughly revised. Italics must be used for plant names.

We have corrected that.

In the Materials and Methods section, a more detailed descriptions are necessary.

More data was provided.

Both an Informed Consent Statement and an Institutional Review Board Statement are necessary due to the involvement of humans in the current study.

Data was provided.

English language and style checking are required.

The paper has been revised by a professional English language interpreter

Round 2

Reviewer 3 Report

The Authors have revised their ms.